# Development of a Mobile Analytical Chemistry Workstation Using a Silicon Electrochromatography Microchip and Capacitively Coupled Contactless Conductivity Detector

**DOI:** 10.3390/mi12030239

**Published:** 2021-02-27

**Authors:** Yineng Wang, Xi Cao, Walter Messina, Anna Hogan, Justina Ugwah, Hanan Alatawi, Ed van Zalen, Eric Moore

**Affiliations:** 1Life Science Interface Group, Tyndall National Institute, University College Cork, T12 R5CP Cork, Ireland; yineng.wang@tyndall.ie (Y.W.); xi.cao@tyndall.ie (X.C.); walter.messina@tyndall.ie (W.M.); Annamaria.hogan@ucc.ie (A.H.); Justina.ugwah@tyndall.ie (J.U.); hanna.alatawi@tyndall.ie (H.A.); 2Chemistry Department, University College Cork, T12 K8AF Cork, Ireland; 3Programme Manager CBRN, Netherlands Forensic Institute, 2490 AA The Hague, The Netherlands; ed.van.zalen@nfi.nl

**Keywords:** microfluidic microchip, capillary electrochromatography, capacitively coupled contactless conductivity detection, microchip electrochromatography

## Abstract

Capillary electrochromatography (CEC) is a separation technique that hybridizes liquid chromatography (LC) and capillary electrophoresis (CE). The selectivity offered by LC stationary phase results in rapid separations, high efficiency, high selectivity, minimal analyte and buffer consumption. Chip-based CE and CEC separation techniques are also gaining interest, as the microchip can provide precise on-chip control over the experiment. Capacitively coupled contactless conductivity detection (C^4^D) offers the contactless electrode configuration, and thus is not in contact with the solutions under investigation. This prevents contamination, so it can be easy to use as well as maintain. This study investigated a chip-based CE/CEC with C^4^D technique, including silicon-based microfluidic device fabrication processes with packaging, design and optimization. It also examined the compatibility of the silicon-based CEC microchip interfaced with C^4^D. In this paper, the authors demonstrated a nanofabrication technique for a novel microchip electrochromatography (MEC) device, whose capability is to be used as a mobile analytical equipment. This research investigated using samples of potassium ions, sodium ions and aspirin (acetylsalicylic acid).

## 1. Introduction

Currently, micropillar array is broadly used for DNA/RNA separation depending on its filtering feature to easily separate large-sized molecules [1,2]. O. Gustafsson and J. P. Kutter [3] indicated that the microfabricated stationary phase has significant separation performance compared to conventional analytical chemistry research. Microchip capillary electrochromatography (CEC) possesses several advantages compared to traditional CEC: (1) the small size makes microchips more suitable for on-site analysis compared to the bulky benchtop instruments used in analytical laboratories [4]; (2) microchips with a shorter length separation channel achieve faster separation and higher electric field strength even with relatively lower applied voltages; (3) the small dimension of the device also reduces sample consumption and operating costs [5]; and (4) it avoids the difficulty of packaging the stationary phase (particles or beads) evenly in a liquid chromatography (LC) column or microfluidic channel, and reduces the energy and time [6]. As early as 1992, W. D. Folkmot and R. H. Austin built a SiO_2_ array to unfold large-scale molecules [7]. CEC microchip separations were demonstrated on both silica and organic polymer-based monoliths [8,9,10]. F. Ringer and his team built the micropillar arrays in a column with microfabrication technique, which significantly increased the surface-to-volume ratio and separation efficiency [11,12,13].

The first capacitively coupled contactless conductivity detection (C^4^D) method was applied in electrophoresis in the 1980s [11,12,13]. Later, this technology developed rapidly in the 1990s with capillary zone electrophoresis (CZE) separation [8]. Chip-based electrokinetic separation techniques were also gaining interest, especially for bioanalytical purposes [9]. C^4^D as a detection technique has an advantage over the conventional lab-based benchtop methods because it is easy to set up and maintain with the lower operating cost. It also offers a contactless electrode feature which prevents contamination of the solution and can be easily operated [8].

The illustrated electrode configuration in Figure 1 was improved from J. A. Fracassi da Silva’s and C. L. do Lago’s equivalent circuit [8,10]. C_1_ to C_n+1_ and R_1_ to R_n_ are the contributions of hypothetical slices of capillary under the electrodes. Resistors R_1_ to R_n_ are from the buffer solution inside the capillary in the area of the electrode cell. The capacitors C_1_ to C_n+1_ are the silica wall thickness and the C_0_ is the stray capacitance between the two electrodes E_1_ and E_2_. C_0′_ and C_0″_ are the capacitance between electrodes to the grounded metal plate. In an alternating current (AC) circuit, the R_e_ is the impedance of the analyte in the region of the electrode cell. The C^4^D electrode cell design is important so as to minimize the C_0_ to obtain better signal-to-noise (S/N) ratio with a wide range of linearity. Brito-Neto et al. introduced the approach to reduce the stray capacitance by using a metal plate between the electrodes to isolate them, for instance, a capillary through a vertical hole on the grounded metal plate [8,10]. This design helps to lower the stray capacitance and obtain a better signal.

The silicon-based microfluidic chip fabrication processes employ photolithographic and microfabrication techniques, to achieve a highly regular array of silicon pillars. The design focuses on the size, the shape, with a standardized changeable position, which allows the separation column to perform continuously [4]. The fabrication of such a column is a challenge. Eijkel in his review of the recent development of microfabricated column chip-based high-performance liquid chromatography (HPLC) reported the successful cases. Eijkel indicated that the applied microfabrication process on substrates has superior performance compared to the general particle-packed monolithic columns [5]. More care is required in the optimization of the design to obtain the uniform surface quality for the high aspect ratio (partly) porous column as an appropriate component to the microchip-based HPLC or CEC separation [9,11]. The separation efficiency with sophisticated fabricated fluidics systems is generally improved when replaced with relatively polydisperse and heterogeneous packing particles using lithographically microfabricated pillars [12]. Recent studies have shown that when nearly perfect ordered pillar arrays are compared to traditional monolithic column systems, the former demonstrate lower flow resistance with decreased pressure demands [13]. The surface coating, which comprises a uniform coating of the microfabricated pillar column, is another advantage that enhances separation performance. Non-uniformity in the coating impacts on the plate height, presenting inconsistencies in the flow path [11]. 

The conduct ratios and the parameters of the simplified van Dimeter equation [14] for both micro-HPLC and CEC have been evaluated. Guves et al. reported that the efficiency of the monolithic columns in CEC was significantly higher than the micro-HPLC with the same conventional particulate packing method [15]. They demonstrated the benefit of the relaxation of band-broadening with the electroosmotic flow (EOF) in a viscous circumstance. Unlike the pressure-driven injection method such as the conventional HPLC, electrokinetic sample injection loads sample into a microchip on the generation of the EOF. Many electrokinetic injection modes are employed to create a fine sample plug [16,17,18,19,20,21]. The flow performed in EOF-driven systems shows a flat plug, rather than the rounded laminar flow profile characteristic of the conventional pressure-driven flow in chromatography columns. The reason is that EOF does not significantly cause the band-broadening as seen in the pressure-driven chromatography [22].

The performance of electrochromatography technique relies on (1) the electroosmotic flow; (2) the analytes’ electrophoretic mobility and the chromatographic retention; and (3) selectivity of analytes between the mobile phase and the stationary phase. Furthermore, electrokinetic injection is dependent on the channel wall surface, and it is influenced by the pH, ionic strength and viscosity of the analytes [23]. There is the important challenge of the Joule heating effects on peak broadening in capillary electrophoresis (CE) caused by passing an electric current through the buffer solution when an axial electrical field is applied [24]. This is usually done to induce EOF and electrophoretic flow; along with the temperature rise, it impacts the local temperature-dependent electrical conductivity, the diffusion coefficient and dynamic viscosity [25,26,27]. 

C^4^D has become a universal detection method coupled with CE or microchip capillary electrophoresis (MCE) over the past two decades [8,10]. This method of conductivity detection was applied to the traditional ion chromatography (IC) system for ions analysis and the conductivity of the eluent mobile phase. Over the last 50 years, there has been a significant advancement in electronic integrated circuits [28] with the possibility to miniaturize and integrate the C^4^D into a portable workstation system [16,29]. The C^4^D electrode cell concept can be designed to accommodate many types of applications as shown by Coltro et al. [17]. The primary challenge still remains the stray capacitance between electrodes which must be at a minimum [10]. Da Silva demonstrated that [18] a quartz capillary through a hole of a thin metal plate with a ground connection can be used to isolate two electrodes to reduce stray capacitance. Mahabadi et al. [19] also investigated a microchip system with grounded metal thin sheets using a similar technique with a more complicated setup. Generally, during higher frequency excitation, the current does not fluctuate and is determined by the solution resistance only. However, the stray capacitance dominates the output signal when excitation frequency increases [20,21]. Stray capacitance could further be reduced by combining the following routines: (1) designing the electrodes to be directly opposite each other with less area for the pickup electrodes; and (2) introducing a capillary through the grounded plane for isolation of the electrodes [8,10,18,19,20]. 

In this study, a simple and effective electrode cell with grounded feature (Figure 2) was designed and fabricated based on the previous research. The challenge was that the stray capacitance dominates the output signal when excitation frequency is high. The performance of C^4^D depends on the geometry of the electrode cell along the capillary or the fluidic channel. The capillary wall capacitances that are in the area of the electrodes should be as high as possible, and the stray capacitance should be as low as possible, according to Brito-Neto et al. [8,10]. A thin wall fluidic channel for detection is preferred. 

Hexagonal monolith structure described by Bing et al. [30] indicated that a higher surface area to volume ratio, greater distance between the mixing nodes and the unswept channels achieved better separation efficiency. Similarly, this research utilized the nanofabricated slot shape pillars to build the microchip separation column. Based on the team’s recent fabrication experience, technique and infrastructure available, 10 µm width for each slot shape pillar was chosen. This helped to improve the column performance and enhance fabrication quality. 

Lin et al. [31] designed and evaluated various injection ports for electrophoresis microfluidics devices to obtain the lowest leakage injection technique. This is because microchip separation and detection require accurate volume control. Sample leakage during the injection or separation might lead to a broader peak and cause interference in the results. Furthermore, increased injection time might raise the signal baseline due to a broader sample distribution [31]. A low-leakage injection feature is the key to improving the total detection efficiency [32]. This useful technique has been applied to many conventional electrophoresis microchips to demonstrate the advantage of a low sample leakage and continuous sample injection features [3,4,25,26,27]. 

This research applied micro/nanofabricated CEC microchip with C^4^D detection techniques embedded in a custom-built analytical chemistry workstation. All the experiments, fabrication process and device packaging methods with discussions are presented.

## 2. Materials and Methods 

### 2.1. The Fabrication Process of Pickup Electrode Cell Glass

The gold pickup electrode cell was fabricated on a 1 mm thick Pyrex 7740 glass wafer substrate, commercially available from University Wafers (Boston, MA, USA). A classic “lift-off” lithography technique was employed to build an undercut shape. First, spin and coat layer of S1813 photoresist was performed, then another spin and second coat layer of LOR10A photoresist on top of S1813 was performed. This resulted in the different develop rate of these two types of photoresist. The developer MF319 removed more materials at S1813 layer than the LOR10A layer. An undercut shape was achieved which facilitated the metal deposition layer to lift off. Electron beam physical vapor deposition (EBPVD) was used to coat a 30 nm titanium, followed by a 200 nm gold. The titanium was first coated to improve the surface adhesion of the gold layer. The electrode cell body was fabricated on the glass wafer substrate. 

### 2.2. Silicon Microchip Fabrication Process

Figure 3 presents the silicon microchip fabrication processes on a 1 mm thick, 100 mm diameter, one-side polished silicon wafer substrate. A thermal oxidation process was employed to create a thin SiO_2_ protective layer on the substrate. A plasma enhanced chemical vapor deposition (PECVD) process was applied to enhance the SiO_2_ protective layer on the unpolished face. A conventional lithography method was employed on the silicon wafer substrate to initially frame all the microfluidic features. A spin and thin layer coat of photoresist (S1813) was applied on the substrate. This was exposed with UV throughout the lithography photomask. The developer (MF319) was applied to remove the UV exposed pattern of the photoresist. A deep reactive ion etching (SPTS-DRIE) technique (Orbotech, Newport, UK) was applied to open the solid SiO_2_ layer on the polished face. An advanced silicon etching (ASE) process was used to etch 40 µm (actual measured 32.90 µm) depth microfluidic features on the polished face of the silicon wafer. The photoresist layer was then removed by organic solution. All the microfluidic features, including the sophisticated separation column, were shaped on this silicon wafer substrate. A layer of 6 µm thick aluminum was sputtered on the etched microfluidic parts. This was done to protect the delicate features from accidental scratches or falls as well as to ensure the etching of all the silicon wafer substrate. In the dry etcher, the chamber under the silicon wafer was filled with helium gas. If the plasma etched through the substrate, the dry etcher would trigger the alarm, and the etching process would stop. The 6 µm thick aluminum layer ensured the plasma etched the entire silicon wafer substrate equally without triggering the alarm.

To shape the reservoirs, the silicon wafer substrate was flipped and SPTS and ASE techniques were applied to etch through the whole 1 mm thick silicon wafer until the plasma reached the aluminum layer. NaOH was applied to remove the aluminum layer. The silicon wafer substrate was then sent for thermal oxidation again to create a thin equal layer (approximately 10 nm) of SiO_2_ on the etched surface. This step was essential to create a fused silica-like microfluidic channel which has the same surface chemistry as the conventional capillary. To seal all the microfluidic features on the substrate, an anodic bonding process was employed. A 10 nm SiO_2_ layer was deposited on 100 μm Pyrex 7740 glass before bonding to create the fused silica-like surface chemical property of the microfluidic channel. Ten nm of SiO_2_ was determined as the layer deposition needed to prevent bond failure as a result of gaps and bubbles between the silicon substrate and Pyrex glass. Once bonding was achieved, the silicon microchip was ready for integration.

### 2.3. Silicon Microchip Assembly

The custom design of the silicon microchip assembly with the computer numerical control (CNC) is shown in Figure 4. The silicon microchip and electrode cell glass were coupled together under pressure. The steel cartridge was machined precisely to provide proper packaging pressure so as to not damage the chip. Four pieces of Nanoport™ connector (purchased from Kinesis Scientific, Saint Neots, UK) were bonded to the rough face of a silicon microchip. This provided a standard fluidic female port to a male Nanotight™ connector (10–32 coned, compatible with 1/16 tubing). These components offered an ideal microfluidic interconnection, especially under a limited space.

### 2.4. Experimental Platform Assembly

The assembled analytical chemistry workstation alternatively named “Suitcase” system consisted of a Surface Pro tablet computer (Microsoft; 1 Microsoft Way, Redmond, WA, USA), the HV (high-voltage) sequencer (Mode HV448-6000D; LabSmith, Livermore, CA, USA) and the C^4^D detector (TraceDec C^4^D detector; Innovative Sensor Technology, IST AG, Ebnat-Kappel, Switzerland). A custom-designed microchip adaptor platform was used to provide an accessible interface to the microchip and primary system. The microchip adaptor platform was located at the right side of the system housing the microchip assembly as well as performing the analysis. Four platinum wires mounted on the microchip adaptor delivered high voltage to drive the EOF inside the fluidic channel. Spring-loaded contacts (2.54 mm Pitch, Spring Probe, model: P25-0822, HARWIN, UK) were used to ensure the electrical connection to the C^4^D pickup electrodes during signals acquisition. The microchip adaptor platform had an interlock switch feature that provided HV safety to the users. There was a 24 V Li-Po battery (Total 155.4 Wh, 7000 mAh 7.4 V, three pieces in series, LRP electronics GmbH, Schorndorf, Germany) connected using an AC–DC converter to feed the Surface pro, the HV sequencer and C^4^D detector. Three cooling fans (Mode DP200A, Sunonwealth Electric Machine Industry Co., Kaohsiung, Taiwan, ROC) with filter provided fresh airflow to cool all devices inside the suitcase. All these components were arranged in a custom-made frame set up in the “Suitcase” (Peli 1620 Waterproof Wheeled Case. Pelican Products, Torrance, CA, USA).

### 2.5. Instrumentation

Figure 5 shows the custom integrated mobile analytical chemistry workstation “Suitcase” system. This workstation was used for all the experiments. 

### 2.6. Experiment Procedures, Tools and Chemicals

A regular 20 mL syringe was modified for loading sample or generating a vacuum. A LuerTight™ adaptor connected the syringe with 1/16 size tubing and was also associated with a male NanoPort™ connector; it finally connected with the NanoPort™ on the microchip assembly. The buffer solution: 12.5 mM MES/His buffer (2-(N-morpholino)-ethanesulfonic acid, histidine; purchased from Sigma-Aldrich, Dublin, Ireland) was prepared. It was then filtered using 0.2 μm regenerated cellulose syringe filter before use. The deionized (DI) water had a resistivity of 18.2 MΩ and was sourced from a Milli-Q (Millipore, Molsheim, France) water purification system. NaCl and aspirin (both purchased from Sigma-Aldrich, Ireland) were prepared with serial dilution in a buffer solution and stored at room temperature. 

### 2.7. Na^+^, K^+^ and Aspirin 

The test samples were 5 mM NaCl and KCl in 12.5 mM MES/His buffer solution and 20 mM aspirin in 12.5 mM MES/His buffer solution. The silicon microchip was preconditioned by connecting a vacuum to the Reservoir C; the solution was then extracted from the top Reservoirs A, B and D. A pipette was used to continuously top-up NaOH, DI water and buffer solution sequentially through the Reservoirs A, B and D. The NaOH flowed through the column and reached the bottom Reservoir C. This process was performed for 10 min per each of the three liquid solutions.

## 3. Results and Discussion

### 3.1. Voltage Program Selection

A primary check was performed to ensure all the microfluidic channels in the silicon chip were not blocked. DI water was injected to flush all the microfluidic channels. Figure 6 demonstrates that the water drops remained in the microfluidic channel and were observed through a charge coupled device (CCD) camera microscope. After that, a preconditioning process was carried out; Reservoir C was connected to a vacuum source, use pipette sequentially added 0.1 mM NaOH, DI water and 12.5 mM buffer solution to each Reservoirs A, B, and D. This step takes 10 minutes for each type of liquid.

Various voltage setup trials were performed; the data reading (momentary subjective reading) was directly obtained from the HV sequencer software interface, which is listed in Table 1. The C^4^D instrument limits its critical current less than 200 μA. It is necessary to measure the current as well as to optimize the HV-driven EOF. This is to ensure a minimum current is circulating through the microchip. If the microchip overheats, the viscosity of the solution changes and the solution in the reservoirs evaporates.

At higher voltage, higher joule heat or ohimc heating was observed, even though the microchip passivated with 10 nm chemical vapor deposition (CVD) coated SiO_2_ was thermally oxidized on the column inner surface. The silicon microchip still exhibited this peculiar “semi” conductor attribute instead of a good insulator property like polymers. Therefore, a low voltage must be used so as not to overload the current and damage the C^4^D sensor. Linear regression was plotted, the linear region (under Reservoir A 40, Reservoir B 60 setup) is an acceptable voltage setup for the C^4^D sensibility. The final selected voltage for the microchip was 45 V for Reservoir A and 60 V for Reservoir B, Reservoir C and D to the ground. As shown in Figure 7, this voltage setup was the optimal voltage that can be used for analysis with our custom-made microchip system. 

### 3.2. Analysis of Na^+^, K^+^ and Aspirin

The test sample was injected in Reservoir A through a pipette. Figure 8 illustrates the injection and separation technique used in this research. 

The primed silicon microchip was mounted onto the custom-designed microchip adaptor platform. The Pt electrodes (platinum wire, 1.0 mm diameter, 99.99% metals basis, Sigma-Aldrich, Ireland) were inserted into each reservoir. Previously determined voltages were applied and data were recorded. The electropherogram of results obtained for the samples is shown in Figure 9 with Table 2 showing the migration time relative abundance (peak area) and peak height. The test was performed three times to ensure the reliability of the results.

As seen in Figure 9, the results presented a noisy baseline and current overload problem. These problems may be a result of the material’s insulation property. It may also be an indication that the SiO_2_ surface layer created by thermal oven baking and CVD was not sufficient to insulate the HV drive current. Thus, the silicon microchip was only able to run with a lower HV program (Reservoir A: 45 V, Reservoir B: 60 V, Reservoir C and D grounded).

## 4. Conclusions 

This research demonstrated a microfluidic chip device fabrication and packaging process. It explored the capability of this nano/microfabricated silicon microchip device with C^4^D detection technique. The preliminary test result indicated this silicon microchip with C^4^D detection method only can be used for separation and detection under lower voltage setup. An unexpected problem of current overload which was seen in results was obtained as voltage versus current linearity response was encountered during the voltage setup trials. This current overload problem of the microchip might be a result of silicon material with its semi-conductor property. The fabrication processes may have created a SiO_2_ insulation layer which was not sufficient to insulate the current throughout the silicon microchip. The effect of this overheating of the system with a higher voltage setup was subjectively observed.

## 5. Future Works

An atomic layer deposition (ALD) process will be used to deposit an equal coating of 100 nm thick SiO_2_ layers over all the microfluidic channels of this silicon microchip. This will make the inner channel surface similar to a conventional quartz glass microchip to reduce the current throughout the silicon material. The reservoirs will be redesigned and larger diameters will be considered. Furthermore, an optical sensing method will be added onto the existing system to implement parallel detection.

## Figures and Tables

**Figure 1 micromachines-12-00239-f001:**
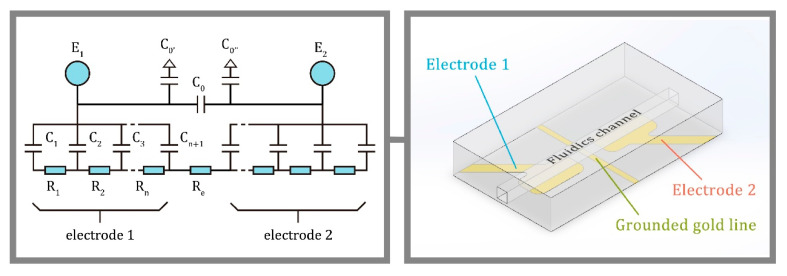
The capacitively coupled contactless conductivity detection (C^4^D) electrode cell (with grounded line feature) equivalent circuit.

**Figure 2 micromachines-12-00239-f002:**
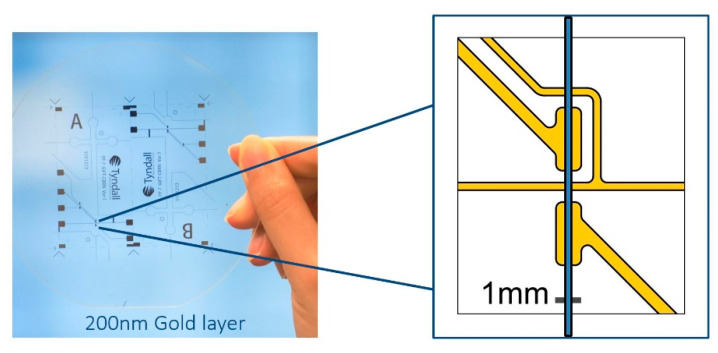
Custom-designed and fabricated pickup electrode cell glass with electron beam physical vapor deposition (EBPVD) coated 200 nm thick and 1 mm width gold electrode cell; a thin grounded line turns around the receiver electrode.

**Figure 3 micromachines-12-00239-f003:**
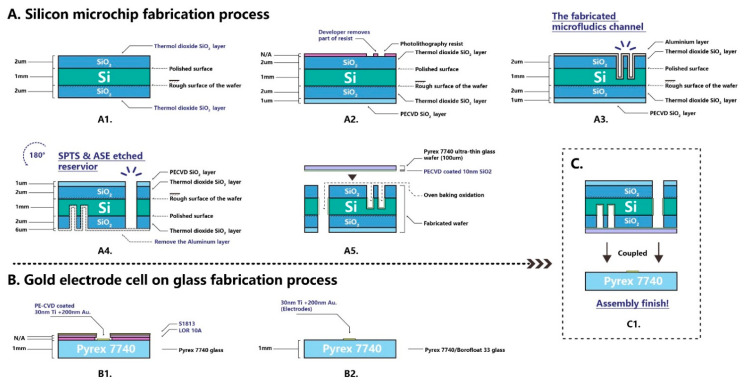
The silicon microchip fabrication process. (**A**) Silicon microchip fabrication process. (**B**) Gold electrode cell fabrication process. (**C**) Packaging of silicon microchip and electrode cell.

**Figure 4 micromachines-12-00239-f004:**
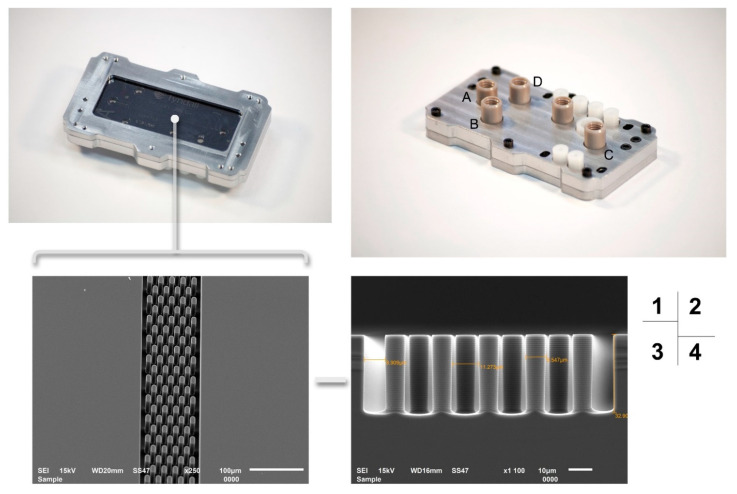
The completed silicon microchip assembly with the steel cartridge. **1:** Microchip assembly with a check window. **2:** Silicon microchip assembly with reservoirs’ name (A, B, C, D). **3:** SEM of nanofabricated column details zoomed from the microchip check window. **4:** SEM of the side view of nanofabricated column details, including the measured size of individual silicon pillar.

**Figure 5 micromachines-12-00239-f005:**
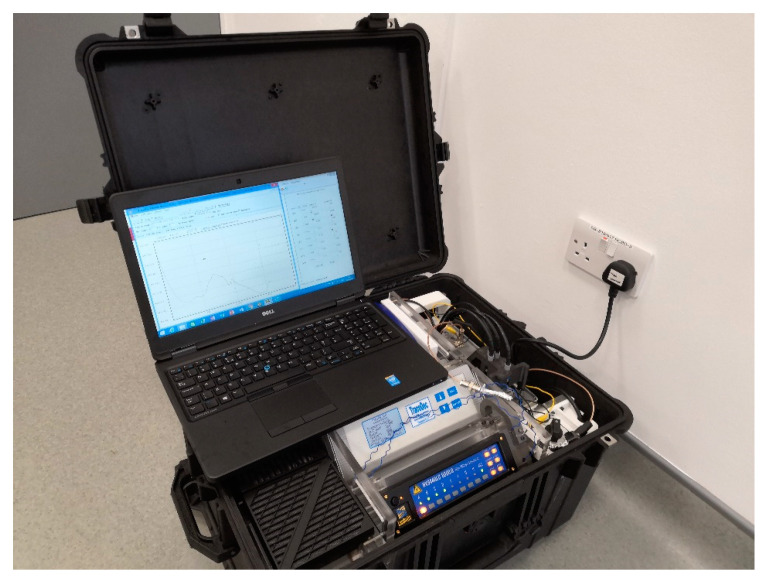
The “Suitcase” system.

**Figure 6 micromachines-12-00239-f006:**
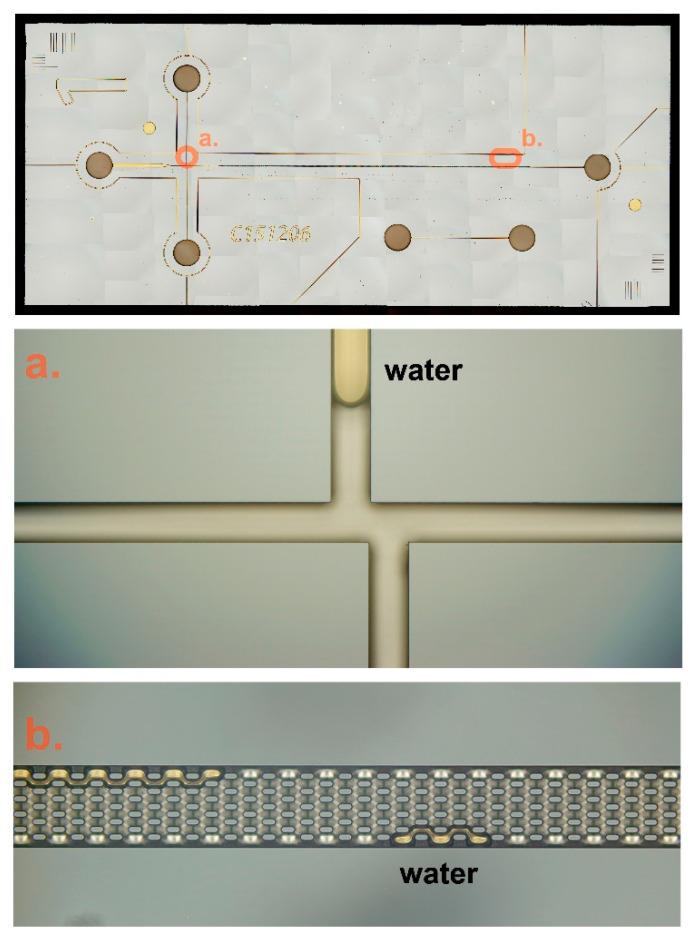
Composite of different microscopic images (52 mm × 22 mm). (**a**) illustrates the double L injection, (**b**) illustrates nanofabricated column. The amber areas are water drops remaining in the fluidics channel.

**Figure 7 micromachines-12-00239-f007:**
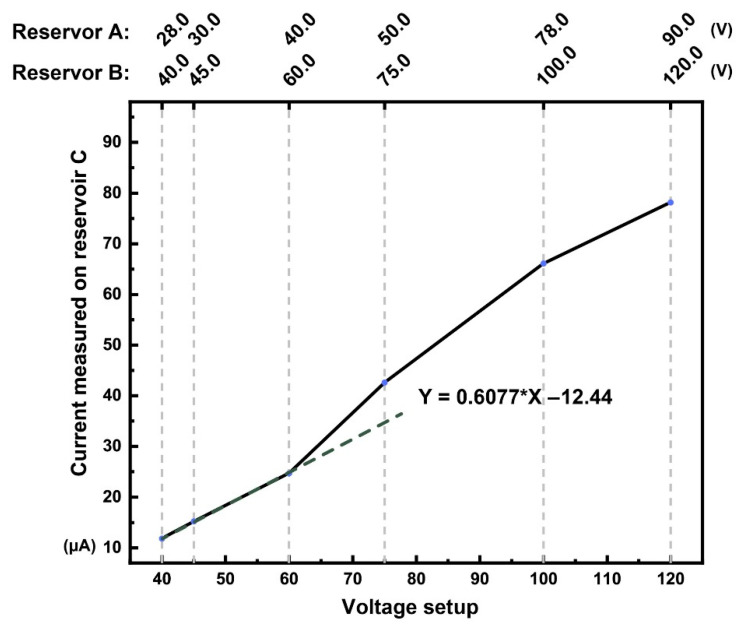
The silicon microchip voltage versus current plot.

**Figure 8 micromachines-12-00239-f008:**
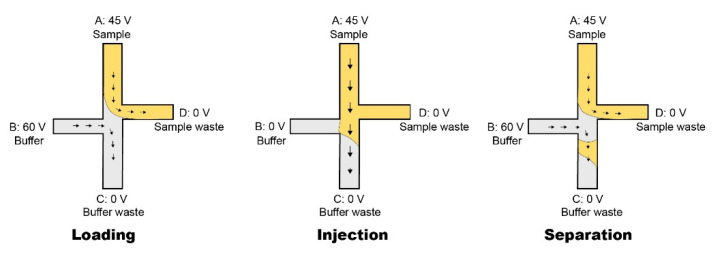
The double-L injection technique was applied to this paper’s research work. **Loading**: the sample was added in the Reservoir A; due to the electroosmotic flow (EOF), it streamed to the Reservoir D. This filled the junction with the sample for injection. Due to voltage difference between Reservoir B and C, the buffer solution was driven from Reservoir B to C. **Injection:** the voltage was only applied to Reservoir A; hence, the sample loaded into the top of the microfluidic column. The injection process was set to a short time (few seconds). **Separation:** the setup was the same as loading, the voltage applied on the Reservoir B was online again. This caused the EOF to drive the buffer solution to push the injected sample plug into the column. After a couple of minutes, the peak appeared on the screen.

**Figure 9 micromachines-12-00239-f009:**
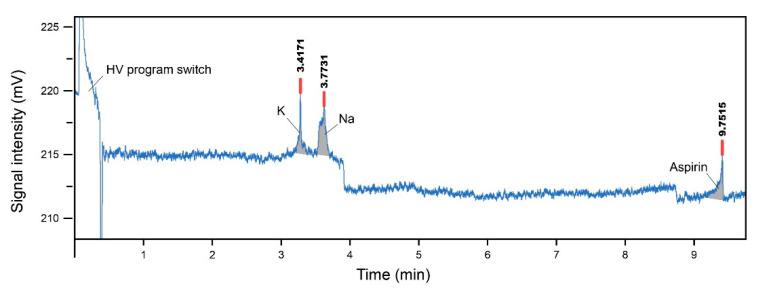
The electropherogram result of the silicon microchip with K^+^, Na^+^ and aspirin, with a running buffer of 12.5 mM MES/histidine.

**Table 1 micromachines-12-00239-t001:** Voltage setup trials for the HV sequencer.

Reservoir	Voltage	Current	Reservoir	Voltage	Current
A	27.387 V	5.74 µA	A	29.257 V	7.91 µA
B	39.112 V	5.98 µA	B	44.723 V	8.23 µA
C	GND *	−11.82 µA	C	GND	−15.25 µA
D	GND	−0.21 µA	D	GND	−0.44 µA
**Reservoir**	**Voltage**	**Current**	**Reservoir**	**Voltage**	**Current**
A	39.782 V	12.21 µA	A	49.571 V	18.43 µA
B	59.811 V	15.74 µA	B	74.311 V	25.47 µA
C	GND	−24.73 µA	C	GND	−42.62 µA
D	GND	−0.98 µA	D	GND	−1.33 µA
**Reservoir**	**Voltage**	**Current**	**Reservoir**	**Voltage**	**Current**
A	89.516 V	40.89 µA	A	119.548 V	30.41 µA
B	119.393 V	43.81 µA	B	159.455 V	80.55 µA
C	GND	−78.19 µA	C	GND	−99.94 µA
D	GND	−0.74 µA	D	GND	−3.46 µA

* GND = grounded.

**Table 2 micromachines-12-00239-t002:** K^+^, Na^+^ and aspirin results.

Sample	Migration Time (min)	Peak Area (A.U.)	Peak Height (mV)
K+	3.4171	3.42	4.06
Na+	3..7731	3.77	3.37
aspirin	9.7515	9.75	2.98

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
