# Peer review of "Development of a Mobile Analytical Chemistry Workstation Using a Silicon Electrochromatography Microchip and Capacitively Coupled Contactless Conductivity Detector"

_micromachines, 2021, doi:10.3390/mi12030239_

Round 1
Reviewer 1 Report
The manuscript dealing with CE/CEC and C4D provides a technically interesting concept but it lacks novelty and the presentation is not adequate. The shown performance is not good and the discussion about the results and problem is not of high quality. The manuscript should be revised by a professional native English speaker. Commenting the language in detail is beyond the duties of a reviewer but my view is that as such the English is not adequate for publishing in a high-quality journal. English proofreading would make the manuscript more readable, clearer, and enhance the message of the author.
Some detailed comments are below
Page 2, line 50-51: Claiming more performance than optical or even MS detection is not supported and, in my view, cannot be supported in case on MS.
Figure 3: text in the figure is too small and almost unreadable. Also, a lot of information is repeated/copied in the figure caption and main text.
Page 5, fab process: How were the reservoirs patterned? Masking for the etching is not described.
Figure 4: C is not a zoomed view but a SEM image. In D the yellow text is too small to read.
Table 1 and Figure 6: The values presented are not valid considering the voltages used later for the measurements. Currents with the final selected values should be measured and discussed. The reservoirs A to D are not explained in reference to the actual chip.
Table 1 and corresponding text: Are the currents average or momentary?
Figure 7: The injection technique is very commonly used and does not need a visual explanation in a research article.
Figure 8 and related text: It is not clear if most of the current goes through the liquid via the channel or through the silicon material. This should be discussed also here, not only in the end. Was the silicon type selected for an HV application? A special low conductivity silicon should be used.
Author Response
Dear reviewer,
A native English speaker has rewritten the manuscript. The gramma, figures, size of fronts etc. from the previous review report was corrected. The details are presented below:
- Page 2, line 50-51: Claiming more performance than optical or even MS detection is not supported and, in my view, cannot be supported in case on MS.
The discuss of MS detection was removed.
- Figure 3: text in the figure is too small and almost unreadable. Also, a lot of information is repeated/copied in the figure caption and main text.
The figure was enlarged and used high-resolution setup. The figure caption information was simplified.
- Page 5, fab process: How were the reservoirs patterned? Masking for the etching is not described.
The reservoirs used our custom-designed CNC machined cartridge to sandwich the reservoirs attach to the microchip. The internal details are relating our intellectual properties; thus, we hope to preserve the design. Regarding the etching masking: we explained in line 165 that the SiO2 protective layer is for the etching masking. The pick-up electrode overall layout was presented in figure 2. The overall microchip layout was shown in figure 6.
- Figure 4: C is not a zoomed view but a SEM image. In D the yellow text is too small to read.
The figure was enlarged and used high-resolution setup.
- Table 1 and Figure 6: The values presented are not valid considering the voltages used later for the measurements. Currents with the final selected values should be measured and discussed. The reservoirs A to D are not explained in reference to the actual chip.
We corrected voltage versus current data range in table 1 for the trials step. The final selected voltage setup was discussed in line 276. The reservoirs A to D were marked corresponding to the actual chip in figure 4-2.
- Table 1 and corresponding text: Are the currents average or momentary?
We all used momentary reading data in this revision, and the problem was corrected in table 1.
- Figure 7: The injection technique is very commonly used and does not need a visual explanation in a research article.
We rewrote the caption, and we hope to keep it for those non-professional audients.
- Figure 8 and related text: It is not clear if most of the current goes through the liquid via the channel or through the silicon material. This should be discussed also here, not only in the end. Was the silicon type selected for an HV application? A special low conductivity silicon should be used.
We investigated the problem of the chip overheating; the voltage versus current plot in figure 7 shows the system lost linearity response beyond the voltage setup of 40 V and 60 V. Thus, the current might flow through the silicon material. We believe the 10 nm SiO2 insulation layer maybe not enough to insulate the higher voltage setup.
Thank you
Best regards
Yineng Wang
Reviewer 2 Report
The authors described a development of mobile analytical work station that utilizes microfluidic chip with coupled contactless conductivity detection technique.
The manuscript described the process of manufacturing the microchip and tested the device on 3 analytes, sodium potassium and aspirin.
I find the manuscript well written.
The data obtained on the use of the device is minimal but authors acknowledged it is preliminary. But maybe the authors can add more data.
The authors mentioned the " the test was duplicated three times to ensure the facticity). Since no statistical analysis was done on the duplicate runs, it would be good to have figure 9 edited to show electropherogram of the other runs. Is the signal response from different concentrations of the analyte expected to be linear?
When testing the device, how many runs could be done before the device fail to function?
What will be the future direction of the authors to solve the issue of noisy baseline and current over load problems?
Author Response

(The authors gave the same response as above.)
